# Management Drives Differences in Nutrient Dynamics in Conventional and Organic Four-Year Crop Rotation Systems

**Sharon L. Weyers [1],* [iD], David W. Archer [2] [iD], Jane M.F. Johnson [1] and Alan R. Wilts [1],†**

[1] USDA Agricultural Research Service, North Central Soil Conservation Research Laboratory, Morris, MN 56267, USA; Jane.M.Johnson@usda.gov (J.M.F.J.); awilts23@gmail.com (A.R.W.)

[2] USDA Agricultural Research Service, Northern Great Plains Research Laboratory, Mandan, ND 58554, USA; David.Archer@usda.gov

\* Correspondence: Sharon.Weyers@usda.gov; Tel.: +1-320-585-8446

† This author has retired.

**Abstract:** Application of exogenous N fertilizers provides agronomic benefits but carries environmental liabilities. Managing benefits and liabilities of N-based fertilizers in conventional (CNV) and organic (ORG) cropping systems might be improved with better knowledge of nutrient dynamics, the generation of intrinsic N, and maintenance of soil organic matter. This study evaluated mineral N dynamics, yields, residue inputs, and change in soil organic C (SOC) and total N (TN) in strip-tilled, four-year crop rotations [corn (*Zea mays* L.)-soybean (*Glycine max* [L.] Merr.)-wheat under-seeded with alfalfa (*Triticum aestivum* L./*Medicago sativa* L.)-alfalfa] over eight years of production under CNV management using mineral-N ($NO_3NH_4$) and chemical pesticides or ORG management using organic-N (animal manure) and no chemical treatments. In ORG, N availability increased over time, but did not benefit ORG yields due to poor control of insects and weeds. Corn, soybean, and wheat grain yields were 1.9 to 2.7 times greater in CNV. In general, SOC was lost in CNV but did not change in ORG. Cumulative yield N removals exceeded cumulative fertilizer-N inputs by an average of 78% in CNV and 64% in ORG. These results indicated ORG management supported N availability by generating intrinsic N.

**Keywords:** N-availability; mineralization; C-sequestration; manure; soil organic matter

## 1. Introduction

Availability and use of N fertilizers, which has increased since the 1960s [1,2], carries environmental liabilities including off-site movement of reactive N that contributes to eutrophication [3]. A recent mass-balance assessment for the upper Mississippi River Basin in the U.S. indicated that in common fertilizer-based management systems fertilizer-N was applied in excess of 35 kg N ha$^{-1}$ yr$^{-1}$ [4]. Additional evidence of excess fertilizer use was gleaned from national survey data [5] that indicated fertilizer application rates often exceeded yield removal rates (Appendix A). Management strategies that can sustain crop yields while reducing excess fertilizer application are needed if negative environmental outcomes are to be avoided or mitigated.

A viable strategy to reduce fertilizer inputs without compromising yields is to increase intrinsic (i.e., internal) soil N sources by increasing soil organic matter (SOM) [6,7]. Increasing SOM stimulates soil biotic activity [8], which promotes N cycling and availability, and reduces N loss [9,10]. However, fertilizer-N sources might influence the ability to increase SOM. Kahn et al. [11] and Mulvaney et al. [12] argue that organic fertilizers (e.g., green or animal manure) increase SOM, but mineral fertilizers increase N availability, which prevents accumulation of SOM [13]. Reid [14] and Powelson et al. [15]

countered this opinion by citing evidence that mineral fertilizers increase SOM because increased crop productivity increases residue (i.e., organic matter) inputs (e.g., [16]). Resolving this issue of fertilizer source impact on SOM dynamics and intrinsic N sources is necessary for determining an effective strategy to reduce external inputs.

The primary goal of this study was to build greater understanding of how management, including fertilizer-N source and application rate, impacts C and N dynamics quantified by N availability and mineralization as relative indicators of intrinsic N sources, and soil organic C (SOC) and total N (TN) as relative indicators of SOM. The study was conducted in strip-tilled (ST), four-year (4y) crop rotation systems under conventional (CNV) and organic (ORG) management, a subset of experimental plots within the Farming Systems Experiment (FSE) first described by Archer et al. [17]. The current study evaluated three years of data on N dynamics in the context of eight years of existing data on microbial C and N, crop yields, residue inputs, and plant and soil C and N balances. The hypotheses tested were (1) greater crop production and residue returns will improve SOM, and (2) improvements in SOM will be associated with increased N availability from intrinsic N sources.

## 2. Materials and Methods

### 2.1. Site Description

The study was conducted within a 3-ha area located on the Swan Lake Research Farm, in Stevens County, near Morris, MN (45°41′ N, 95°48′ W). This study area is representative of the Upper Midwest portion of the US Corn Belt and falls within the prairie parkland ecoregion, a province dominated by Mollisols and a cold but temperate humid climate [18,19]. Experimental plot layout statistically normalized landscape variability in soil properties due to the five soil series present [20]. These soils formed within glacial till under prairie vegetation, which often results in higher soil organic C. They mapped as: Barnes loam (fine-loamy, mixed, superactive, frigid Calcic Hapludoll); Flom silty clay loam (fine-loamy, mixed, superactive, frigid Typic Endoaquoll); Hamerly clay loam (fine-loamy, mixed, superactive, frigid Aeric Calciaquoll); Parnell silty clay loam (fine, semectitic, frigid Vertic Argiaquoll); and Vallers silty clay loam (fine-loamy, mixed, superactive, frigid Typic Calciaquoll) [21]. Surface soil (0–15 cm) properties averaged 33 g kg$^{-1}$ total C, 31 g kg$^{-1}$ organic C, 3 g kg$^{-1}$ total N, 13 mg kg$^{-1}$ P, 182 mg kg$^{-1}$ K, 7.8 pH$_{H2O}$, 36% sand, 27% silt, and 37% clay.

A fully automated weather station at the site collects hourly rainfall and soil temperatures at 5 cm increments to a depth of 15 cm (https://www.ars.usda.gov/midwest-area/morris-mn/soil-management-research/docs/weather/). Additional historical climate data for the region (30 yr from 1981 to 2010) indicated average annual precipitation was 672 mm, air temperatures ranged from a daily average of −12.2 °C in January to 21.4 °C in July, and annual growing degree days averaged 1314 for 10 °C base [22]. For the eight years of the FSE (2002 to 2009), average annual precipitation was 586 mm, with a growing-season average of 401 mm, an average annual temperature of 6.3 °C, and average annual growing degrees days of 1356 (10 °C base). Cumulative precipitation and soil temperature degree days (0–5 cm depth, calculated using a 10 °C base as a temperature that inhibits most microbial activity; [23]) from early April to October for the three years soil mineral N and mineralization were monitored are shown in Figure 1.

### 2.2. Experimental Design

The Farming Systems Experiment (FSE), first described by Archer et al. [17], was established in 2002 on historically cropped ground (>50 yr) last cropped to soybean (2001) in a long-term corn-soybean-spring wheat rotation managed with disk tillage and standard crop inputs. The FSE was laid out as a randomized complete block design with four replicated blocks divided by a split-plot factor, with either half randomly assigned to one of two system-level treatment factors (ORG, organic; or CNV, conventional). Each system contained a full factorial arrangement of three, two-factor management strategy treatments: (1) tillage (CT, conventional; or ST, strip-till); (2) fertility (YF, yes-fertilized; or

NF, not-fertilized); and (3) rotation (2y, two-year; or 4y, four-year) [17]. Nested within rotation was an entry point (EP) treatment factor, producing a total of six rotation-entry point levels. These levels were defined by the first crop in the rotation sequence that was planted in 2002 and were designated: 2y-EP1, corn; 2y-EP2, soybean; 4y-EP1, corn; 4y-EP2, soybean; 4y-EP3, wheat/alfalfa; and 4y-EP4, alfalfa. This rotation-entry point factor established that all crop phases in each rotation were grown every year. Thus, a total of 192 experimental plots (6 × 12 m) were established (4 blocks × 2 systems × 2 tillage × 2 fertility × 6 rotation-entry points). The current evaluation focused on the comparison of ORG and CNV systems, within and across all four entry point treatments in the 32 experimental plot-subset under ST, YF, and 4y rotation strategies (4 blocks × 2 systems × 1 tillage × 1 fertility × 4 rotation-entry points) (Figure 2). A total of eight specific treatment combinations, with four replicates each, were compared and designated here as: ORG-EP1, ORG-EP2, ORG-EP3, ORG-EP4, CNV-EP1, CNV-EP2, CNV-EP3, and CNV-EP4.

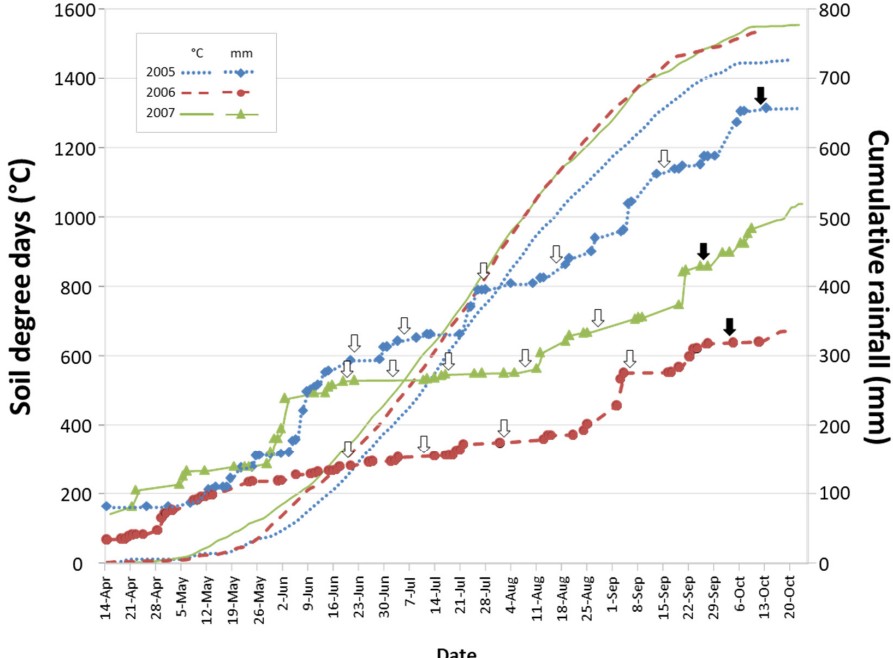

**Figure 1.** Soil degree days (10 °C base), cumulative precipitation, and soil mineralization incubation sampling intervals within the growing seasons of 2005 through 2007. Open arrows indicate initial bulk sampling and starting points for incubation intervals, closed arrows indicate the last sample date for core and resin retrieval. Each rainfall data point represents a single rain event.

*2.3. Management*

Management practices for the ORG system followed standards established under the USDA National Organic Program [24]; this management satisfied the requirement for full certification and application of production premiums by the 2004 harvest season [17,25]. Defined at the system treatment level, management for the ORG system included animal manure amendments as the only external fertilizer source, and the use of only approved or non-prohibited substances as provisioned by the National List (https://www.ams.usda.gov/rules-regulations/organic/national-list). Conversely, system level management in the CNV entailed the use of synthetic substances including mineral fertilizers, insecticides, and herbicides. As required for certification, a minimum 12-m wide buffer area was maintained between the ORG and CNV management system splits within each block and between blocks to avoid synthetic chemical drift into the organic management area.

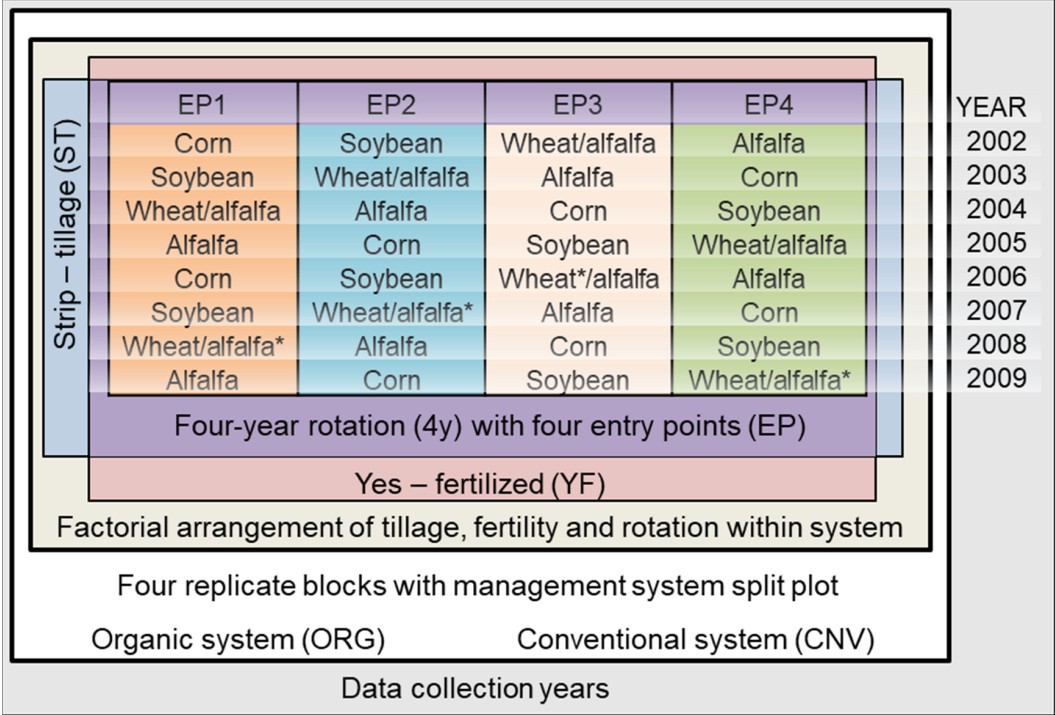

**Figure 2.** Randomized complete split-block experimental plot design emphasizing entry point (EP) factor within the 4y rotation treatment. Each of the four entry points (EP1, EP2, EP3, and EP4) represents a four-year crop rotation sequence that was repeated twice over eight years (2002–2005 and 2006–2009). Outer most gray box shows data collection years aligned with rotation sequence of crops. Inner white box indicates one of four replicate blocks, split by management system. Next inside, the off-white box represents the factorial arrangement of rotation (purple box), strip-tillage (blue box), and yes-fertilized (pink box) treatment factors, that were established in both management systems. Thus, all four EP treatments within both organic (ORG) and conventional (CNV) splits in each of four replicate blocks yielded 32 experimental units. * Wheat in ORG-EP3 was not under-seeded with alfalfa in 2006; alfalfa only was planted in ORG-EP2, ORG-EP1, and ORG-EP4, respectively, from 2007 to 2009.

In the current study, fall strip-tillage was used after corn and alfalfa, and no-tillage after soybean (CNV) and wheat due to under-seeding with alfalfa. In ORG soybean, tillage strategies varied with weed management needs, and included fall or spring strip-tillage, no-tillage, or chisel tillage. Inorganic fertilizer was used in the CNV and organic manures in the ORG. The 4y-rotation treatments were a sequence of corn–soybean–wheat/alfalfa–alfalfa, with the starting point of the sequence defined by entry point. In the ORG, the corn and soybean varieties used were certified organically grown seed; however, wheat and alfalfa were the same varieties used in the conventional system, but the seed was untreated, GMO-free, and certified for organic use. Important management aspects specific to the current three-year evaluation are listed in Table 1 and include crop varieties used, and dates for planting, tillage, insect pest and weed management, and harvest; more specific details for all years of the FSE are provided in Weyers et al. [26,27].

**Table 1.** Dates for tillage, planting, pest and weed control, and harvest by conventional (CNV) and organic (ORG) management system and crop.

| System | Crop | 2005 | 2006 | 2007 | Notes |
|---|---|---|---|---|---|
| | | | **Tillage** | | |
| CNV & ORG | Corn | 2-Nov | 1-Nov | | Strip-Till (ST) |
| ORG | Corn | 29-Apr | 7, 15-May | 6-Jun; (10-May) | Field cultivation (chisel) |
| CNV & ORG | Soybean | 12-May | 22-May | 14-May | ST-CNV, Chisel-ORG |
| CNV | Alfalfa | 2-Nov | 1-Nov | | ST |
| ORG | Alfalfa | 3-Nov | 6-Sep | | Chisel |
| | | | **Planting** | | |
| CNV | Corn | 29-Apr | 27-Apr | 3-May | DK4446 |
| | Soybean | 10-May | 16-May | 9-May | Pioneer 91B33 RR plus |
| | Wheat | 28-Apr | 27-Apr | 27-Apr | Oakley (2005); Alsen |
| | Alfalfa | 28-Apr | 27-Apr | 3-May | Wrangler |
| ORG | Corn | 29-Apr | 16-May | 6-Jun | NC + 0M21 (2005); Blue River 30A-12 |
| | Soybean | 10-May | 30-May | 6-Jun | Vital |
| | Wheat | 28-Apr | 27-Apr | not planted | Oakley (2005); Alsen |
| | Alfalfa | 20-May | not planted | 6-Jun; 4-May | Wrangler; Big N (4-May-07) |
| | | | **Insect control**[†] | | |
| CNV | Soybean | 29-Jul | 20-Jul | 10-Aug | Warrior (aphid control) |
| | | | **Weed control** [†] | | |
| CNV | Corn | 5-May [‡]; 6, 27 -Jun, | 8 [‡], 31-May; 20-Jun | 11 [‡], 23-May; 20-Jun | Glyphosate (+Clarity [‡]) |
| | Soybean | 6, 27-Jun | 18, 31-May; 20-Jun | 23-May; 20-Jun | Glyphosate |
| | Wheat | | 7-Jun; (14-Aug) | 8-Jun | Buctril (& Poast) |
| | Alfalfa | 11-Sep | 22-Aug | | Glyphosate, termination |
| ORG | Corn | 16, 20, 23, 31-May; (27-Jun) | 7, 28-May; 2 -Jun | 17, 22-Jun | Rotary hoe (& row cultivation) |
| | Corn & Soybean | 27 or 31-May | 31-May; 8, 20, 27-Jun; 3, 12, 24-Jul; 11, 29-Aug | 5, 12, 18-Jul | Mowing |
| | Soybean | 16, 23-May; (27-Jun) | 15, 28-May; 20-Jun | 3, 9-May; 17, 22-Jun | Rotary hoe (row cultivation) |
| | Soybean | 19-May | | | Acetic acid |
| | Wheat | 16, 20-May (31-May) | 15, 28-May; 2-Jun; (8-Aug) | | Harrowing (& Mowing) |
| | | | **Harvest** | | |
| | Alfalfa | 25-Jun | 2-Jun | 6-Jun | 1st cutting |
| CNV & ORG | | 21-Jul | 5-Jul | 5-Jul | 2nd cutting |
| | | 31-Aug | 8-Aug | 7-Aug | 3rd cutting |
| CNV & ORG | Wheat | 4-Aug | 31-Jul | 31-Jul | |
| CNV | Soybean | 23-Sep | 21-Sep | 24-Sep | |
| ORG | Soybean | 23-Sep | 2-Oct | 12-Oct | |
| CNV | Corn | 15-Oct | 10-Oct | 2-Oct | |
| ORG | Corn | 15-Oct | 10-Oct | 24-Oct | |

[†] Active ingredients: Buctril = bromoxynil; Clarity = Diglycolamine salt of 3,6-dichloro-Q-anisic acid; Glyphosate = N-(phosphonomethyl)glycine; Poast = sethoxydim; Warrior = Lambda-cyhalothrin $\alpha$-cyano-3-phenoxybenzyl 3-(2-Chloro-3,3,3-trifluoroprop-1-enyl)-2, 2-dimethylcyclopropane-carboxylate. [‡] Dates for weed control using a mixture of glyphosate and Clarity.

Net inputs of inorganic and organic fertilizer-based mineral N, total N, P, K, and total organic C varied by entry point because plots received crop-specific application rates over the rotation sequence (Table 2). For CNV systems, standard applications were made for corn, soybean and wheat. Corn received a pre-plant fertilizer application of 11 kg N ha$^{-1}$ and 38 kg P ha$^{-1}$ in the forms of ammonium nitrate ($NH_4NO_3$) and diammonium phosphate (DAP, $(NH_4)_2HPO_4$) every year. Corn received a side-dress fertilizer application of 175 kg N ha$^{-1}$ ($NH_4NO_3$) once in 2002 (CNV-EP1 treatments, *n* = 4). Soybean received the same amount of pre-plant fertilizer from 2002 through 2005. Wheat received an application of 78, 34, and 34 kg ha$^{-1}$ of available N, P, and K respectively, in the form of $NH_4NO_3$, DAP, and potash every year.

**Table 2.** Fertilizer N inputs by system, entry point and component (mineral N, total N) over the experiment, 2002–2009.

| | CNV | | | | ORG | | | |
|---|---|---|---|---|---|---|---|---|
| | EP1 | EP2 | EP3 | EP4 | EP1 | EP2 | EP3 | EP4 |
| | kg mineral N ha$^{-1}$ | | | | | | | |
| 2002 | 11 + 175 † | 11 | 78 | | 24 | | | |
| 2003 | 11 | 78 | | 11 | | na ‡ | | |
| 2004 | 78 | | 11 | 11 | 52 | | | |
| 2005 | | 11 | 11 | 78 | | | | 56 |
| 2006 | 11 | | 78 | | | | 101 | |
| 2007 | | 78 | | 11 | | | | |
| 2008 | 78 | | | 11 | | | | |
| 2009 | | 11 | | 78 | | | | |
| Cumulative Mineral N | 364 | 189 | 189 | 189 | 76 | na | 101 | 56 |
| Total N (kg ha$^{-1}$) | 364 | 189 | 189 | 189 | 353 | 125 | 403 | 182 |
| Mineralizable N (25–35% of total organic N §) (kg ha$^{-1}$) | | | | | 69–97 § | na | 76–106 | 47–66 |
| Total C (Mg ha$^{-1}$) | | | | | 1.8 + 4.1 ¶ | 4.2 | 7.3 | 4.1 |

† Second value indicates the N fertilizer side dress applied in this first year of corn production; ‡ na indicates data "not available"; § Values represent the 25%–35% range of mineral N potentially mineralized from total organic N (total N minus inorganic N) in the first year of application; ¶ The first and second values refer to each of the fertilizer applications in 2002 and in 2004.

Manure application rates were irregular due to variable moisture and composition. Application rates were based on preliminary estimates from prior years' nutrient analyses and selected to balance P application with crop requirements. Due to this procedure nutrient loads were expected to differ between CNV and ORG. Composted dairy manure was applied in 2002, 2004, 2005, and 2006 at an average application rate of 32.9 Mg ha$^{-1}$. Sampling issues prevented analysis of the dairy manures used in spring of 2002 and 2004. However, five available analyses indicated composition averaged 60% moisture, 6.5 g total N kg$^{-1}$, 5.1 g organic N kg$^{-1}$, 1.4 g inorganic N ($NO_3$ + $NH_4$) kg$^{-1}$, 1.8 g $P_2O_5$-P kg$^{-1}$, 3.7 g $K_2$O-K kg$^{-1}$ and 110 g total C kg$^{-1}$. Liquid hog manure was applied once in late October of 2002 to plots rotating from soybean into wheat for the 2003 growing season, which fell under the ORG-EP2 treatment. This liquid hog manure supplied 125, 29, and 124 kg ha$^{-1}$ total N, P, and K, respectively. Fertilization with manure ceased in 2007 because wheat was removed from the rotation, thereafter alfalfa became the only source of N. Distribution of mineral N and cumulative fertilizer nutrient inputs by entry point through 2009 are listed in Table 2. More specific details of fertilization history are provided by Weyers et al. [26,27].

*2.4. In Situ Mineralization Assessment*

The novel aspect of this current evaluation was the determination of mineral N dynamics during three successive growing seasons from 2005 through 2007 within the surface 0–10 cm. This evaluation was conducted all three years within the eight treatment combinations for the ORG and CNV systems, under ST, YF, and 4y rotation by entry points. The evaluations began each season only after all crops had been planted and all field activities that might have disturbed incubating cores, such as tillage, roto-tilling, or harrowing, were completed for the season. Net N mineralization (Nmin) was determined using the open core resin bag incubation method [28–31]. Briefly, at the start of each sequential incubation period an initial bulk soil sample, as a composite of three 2.5-cm dia cores to 10-cm depth, was taken for determination of soil mineral N (SMN). At this time, a set of three soil cores (using 5-cm dia. by 10-cm length aluminum cylinders) were extracted, the bottom 1 cm excavated and a mesh bag containing 15 g of a 50:50 mix of anionic and cationic resins (Ionac ASB1-P and Ionac C-249, respectively, Sybron Chemicals Inc., Birmingham, NJ) was attached before placing the cores back in the ground for the duration of the incubation period. Sequential sets of cores were incubated over a graduated time course of two, three, or four weeks until the last crop was harvested (Figure 1). Timing

of incubations was determined by frequency of precipitation events, meaning shorter periods under frequent rainfall, longer during dry periods, and delays as needed to allow for soil water drainage following a rainfall event to prevent soil compaction. For 2005, 2006, and 2007, respectively, total days of incubation were 111, 105 and 97 days, set out over 5, 4 and 5 sequential events as indicated in Figure 1.

For initial SMN and incubated N mineralization samples, field-moist soil was passed through a 0.45-cm mesh and triplicate subsamples were extracted with 1 M KCl at a 1:10 soil to solution ratio. Soil weights were corrected to a 105 °C dry mass. The mineral N in resin bags was extracted successively 5 times with 1 M KCl and pooled for analysis. All KCl extracts were analyzed on an Alpkem continuous flow analyzer (OI Analytical, College Station, TX) for total extractable mineral N, measured as $NH_4^+$-N and $NO_3^-$-N + $NO_2^-$-N, following standard colorimetric methods [32]. Soil bulk densities were used to convert extractable mineral N measured in initial and incubated soil (mg kg$^{-1}$), and resin bag mineral N (mg L$^{-1}$) into kg ha$^{-1}$.

*2.5. Supporting Analyses*

Crop yield, plant, and soil analyses conducted on the FSE were previously published [26,27,33–36]. Here the data were assembled for the 32 ORG and CNV experimental plots under the ST, YF, 4y-rotation treatments and statistically reassessed with respect to the entry point treatment factor. Crop grain and forage yields were obtained every year using a two-row plot combine; a subsample of harvested material was evaluated for C and N content from 2002 through 2007 [26,27], except alfalfa which was not harvested in 2002 (EP4). Biomass and nutrient content of vegetative shoot (stover, stubble, or straw) and root material were evaluated from 2004 through 2007 [33,34]. Microbial biomass C and N (MBC and MBN) was evaluated every spring from 2003 through 2010, in every plot to a 15-cm depth [36]. Soil organic C and TN, calculated on equivalent soil mass basis to a maximum 60-cm integrated depth, were obtained from samples taken in the fall of 2002, 2003, 2007, and 2009 in every plot [35]. A synopsis of sampling and analysis methods are provided in Appendix B.

*2.6. Calculations*

For each incubation period, mineralized N was calculated as the amount of total mineral N ($NO_3^-$-N + $NH_4^+$-N) contained in the initial bulk soil sampled prior to incubation (soil mineral N; SMN) subtracted from the total amount of mineral N in the soil that accumulated during the incubation (N incubated) plus the mineral N that was trapped in the resin (N resin) as follows:

$$\text{Mineralized N} = (\text{N incubated} + \text{N resin}) - \text{SMN}. \tag{1}$$

Net N mineralized (Nmin), in kg ha$^{-1}$, was then calculated as the sum of mineralized N over all incubation periods during each growing season.

To allow for a statistical comparison of yields across all crops, corn yield equivalent weight (CYE) was calculated [37] as the gross estimated dollar sales of crop A based on yield and dollar value Mg$^{-1}$ of soybean, wheat, or alfalfa, divided by corn price per unit weight (US$111 Mg$^{-1}$) as follows:

$$\text{CYE Mg ha}^{-1} = (\text{yield Mg crop A ha}^{-1} \times \text{US\$ Mg}^{-1} \text{ crop A})/\text{US\$111 Mg}^{-1} \text{ corn}, \tag{2}$$

where US$ Mg$^{-1}$ crop A was $278 for soybean, $175 for spring wheat, and $102 for alfalfa hay. These values were obtained as average (2002–2009) conventional crop grain and forage prices reported in the National Agricultural Statistics Service Quick Stats Application (https://quickstats.nass.usda.gov/). This approach allowed relative yields of grain and forage across crops and across management system to be compared and is not intended as an economic analysis of these production systems.

Plant data collected in 2004 through 2007 were used to calculate harvest index (HI) and root:shoot ratios for each crop and management system separately. HI was calculated as ratio of grain weight to total weight of above ground biomass including vegetative shoot and grain,

$$\text{HI = grain weight ha}^{-1}/\text{total above ground biomass ha}^{-1}. \tag{3}$$

Root:shoot ratios were calculated as weight of root material to weight of vegetative shoot material,

$$\text{Root:shoot = root weight ha}^{-1}/\text{shoot weight ha}^{-1}. \tag{4}$$

Mass and nutrient content of retained residues was calculated as the sum of actual or estimated vegetative shoot and root residues on an area basis. Vegetative shoot and root materials were estimated for 2002, 2003, 2008, and 2009 using calculated HI and root:shoot ratio and actual crop yield,

$$\text{Estimated shoot residues = crop yield ha}^{-1} \times ((1/\text{HI}) - 1)), \text{ and} \tag{5}$$

$$\text{Estimated root residues = shoot estimate ha}^{-1} \times \text{root:shoot} \tag{6}$$

Shoot and root biomass and nutrient content of alfalfa for 2002 was estimated from average forage production and N contents with the assumption 5% of forage production remained in the field [31]. Shoot and root N content was calculated from known N concentrations, or estimated for missing years, using the average of N concentration for each crop and system from 2004 through 2007.

A net N balance was calculated for each system entry point treatment by subtracting the cumulative amount of N removed as grain or forage yields from the cumulative total N of inputs,

$$\text{Net N balance = cumulative total N inputs} - \text{cumulative yield N.} \tag{7}$$

This calculation does not account for any N that was added to the system through fixation, or any N that might have been lost through other processes such as denitrification or leaching. This also does not account for changes in soil or residue N pools.

## 2.7. Statistical Analysis

All statistical tests used a mixed model procedure (PROC GLIMMIX, SAS 9.4 statistical software; [38]), where non-normal data were tested against a log-normal distribution. Two random effects used in all models were the block effect and the block by system interaction, which accounted for the split-plot structure of the experimental design. CYE and SMN were evaluated with main treatment factors of system, entry point, and year as fixed effects. For SMN, sampling points within growing seasons were modeled as repeated measures. Crop effects were evaluated by treating year and entry point as random factors. Calculated residue mass, C and N returned, and cumulative CYE summed over 2002 to 2009 within each experimental plot, were evaluated for system and entry point fixed effects. Plant biomass and nutrient content of grain, shoots and roots, HI, and root:shoot ratios were evaluated for treatment differences within a crop, with system as a fixed effect, and entry point and year as random effects. For Nmin, system and entry points were evaluated as fixed effects with year as a random effect, due to the different total number of incubation days each year. A second set of models evaluated system, entry point, and system by entry point interaction effects for each year separately. To evaluate change in Nmin over time, Nmin-rate (N mineralized in kg ha$^{-1}$ d$^{-1}$) was determined within each year for each system across entry points, and for each entry point within system, using a regression approach in JMP 13 [39]. Within years, rates were obtained as the slope of cumulative sum of mineralized N (y) by days of incubation (x) fit to a linear regression. All rate slopes were significantly different from zero ($p < 0.001$) and r$^2$ values were above 0.64.

Multiple pairwise comparisons were made using the pdiff function in SAS for all main effects and interactions between system and entry points, using the slice function to isolate comparisons

within years as necessary. Bonferroni adjustments of *p*-values were used when necessary to determine significant differences in planned comparisons. Where necessary, log-normal least square means (LSM) ± log-normal standard error of the mean (SEM) were back-transformed as $e^{LSM}$ and $e^{LSM \pm SEM}$. Due to skew of the SEM when back-transforming from the log-normal, the positive and negative standard errors were averaged to simplify data presentation in text and tables. For all tests, significance was established with $\alpha = 0.05$.

## 3. Results and Discussion

### 3.1. Fertilizer Inputs, Soil Mineral N (SMN), and Yield Response

Summed from 2002 through 2009, fertilizer-based mineral N inputs were greater in CNV than ORG management systems, whereas total N inputs (mineral N plus organic N) were greater in ORG than CNV (Table 2). The difference in mineral N applied to CNV and ORG was expected given that manure application rate was based on prior years' nutrient quality and predicted N availability [40] because nutrient content was not measurable prior to application. Based on current recommendations, available N (i.e., soil mineral N) as a percentage of total organic N supplied was expected to be 25%–35% in the first year of manure application, 12%–25% the second year and less than 5% the third [40,41]. Mineralization of the average 235 kg ha$^{-1}$ organic N in dairy manure at 35%, 25% and 5%, respectfully, could release 82, 38, and 5.7 kg mineral N ha$^{-1}$. Thus, three years would be necessary for mineral N inputs in ORG to meet or exceed the 189 kg N ha$^{-1}$ applied to CNV in a single year, excluding the 2002 corn side dress. This assessment indicated that potential mineral N availability, if just accounting for fertilizer application, would be greater in the CNV than ORG.

Despite the differences in fertilizer source and application rate, SMN, an established index of plant available N [42], averaged across years and entry points, was similar between the CNV and ORG management systems, at 6.5 ± 0.47 and 7.0 ± 0.50 kg N ha$^{-1}$, respectively; however significant differences between systems were dependent on a significant interaction with entry point and year ($p < 0.0001$; Figure 3). Multiple comparisons indicated that SMN was greater in ORG-EP2 than CNV-EP2 in 2005, greater in both ORG-EP1 and CNV-EP1 than all other treatments in 2006, and greater in ORG-EP4 than all other treatments in 2007. Across years, SMN was greater in CNV-EP1, ORG-EP1, and ORG-EP4 (9.1 ± 0.79, 8.8 ± 0.77, 8.6 ± 0.75 kg ha$^{-1}$, respectively) than all other entry points (which ranged from 5.6 ± 0.48 to 6.1 ± 0.53 g ha$^{-1}$). These findings point to potential crop effects whereby SMN (year random) was significantly greater in ORG corn phases than CNV and ORG wheat and alfalfa phases.

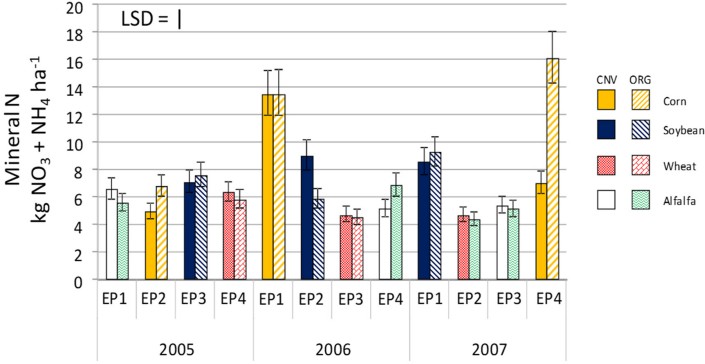

**Figure 3.** Average standing stocks of available soil mineral N (kg NO$_3^-$-N + NH$_4^+$-N ha$^{-1}$) (0–9 cm depth) for each crop by initial crop rotation entry point (EP) for conventional (CNV) and organic (ORG) management systems for the 2005 through 2007 growing seasons. Note: CNV wheat was under-seeded with alfalfa in all years; ORG wheat was under-seeded with alfalfa only in 2005, then dropped from the rotation in 2007. Bars show standard error of the mean. Least significant difference (LSD) is indicated by the dark bar in the upper left corner. Each bar represents the mean of four replicates. Crop rotation EP is defined in Figure 2.

These SMN dynamics did not reflect a direct impact on N availability. One reason for this in the fertilizer application year is that SMN measurements were initiated 40 days or more after fertilizer application. Moreover, wheat had reached anthesis by or just after soil sampling began, but the alfalfa under-seeded into wheat was actively growing. Another reason is that SMN was assessed at 0–10 cm rather than the full depth to which roots might reach. The low SMN content measured during the wheat/alfalfa and alfalfa cropping phases might be explained by the fact that alfalfa has a high N requirement and is particularly effective at removing $NO_3$-N, regardless of active N-fixation by symbiotic rhizobia [43]. The lower SMN observed in CNV corn over ORG corn phases in 2005 and 2007 (respectively, EP2 and EP4) might be explained by greater N uptake capacity of the CNV corn. These data agree with observations made by Kramer et al. [44] of higher uptake rates of 8 kg N ha$^{-1}$ day$^{-1}$ and nutrient use efficiency of 41% of a conventionally managed corn crop compared to the 5 kg N ha$^{-1}$ day$^{-1}$ and 19% efficiency measured for an organically managed corn crop. The absence of this difference in 2006 was probably due to lower rainfall (Figure 1) limiting crop growth and N uptake in both systems.

Averaged over entry points from 2002 to 2009, annual CYE was greater in CNV than ORG systems, respectively, 6.6 ± 0.27 and 3.9 ± 0.28 Mg CYE ha$^{-1}$ yr$^{-1}$ ($p < 0.001$) (Table 3). Within systems, annual average CYE was similar among all CNV entry points, but was greater in ORG-EP1 than ORG-EP2 and ORG-EP3. Partitioned by crop, average CYE of corn, soybean, wheat and alfalfa were significantly greater in CNV than ORG. The cumulative mass of CYE from 2002 to 2009 was similar among all CNV entry points but was significantly greater in ORG-EP1 than all other ORG entry points (Table 3). Among systems and entry points, cumulative CYE was similar among CNV-EP3, CNV-EP4, and ORG-EP1. Over the eight years of production the only trend for changing yields was a significant increase in ORG corn yield of 0.6 ± 0.5 to 5.8 ± 0.5 Mg ha$^{-1}$ yr$^{-1}$ from 2003 to 2007 ($p < 0.001$) (Figure S1).

**Table 3.** Annual and cumulative crop yields in corn yield equivalents (CYE) by entry point and system.

| | EP1 | EP2 | EP3 | EP4 | Avg | Corn | Soybean | Wheat | Alfalfa |
|---|---|---|---|---|---|---|---|---|---|
| Annual | | | | | Mg CYE ha$^{-1}$ yr$^{-1}$ | | | | |
| CNV | 7.0 [a,†] | 6.7 [a] | 6.5 [a] | 6.2 [a] | 6.6 [A] | 8.0 [a] | 6.4 [b,c] | 4.5 [d] | 7.6 [a] |
| ORG | 4.9 [b] | 3.3 [c] | 3.3 [c] | 3.8 [b,c] | 3.9 [B] | 3.5 [d,e] | 3.1 [e] | 2.8 [e] | 6.3 [c] |
| Cumulative | | | | | Mg CYE ha$^{-1}$ | | | | |
| CNV | 56 [a] | 53 [a] | 50 [a,b] | 45 [a,b] | 51 [A] | | | | |
| ORG | 39 [b] | 27 [c] | 26 [c] | 27 [c] | 30 [B] | | | | |

[†] Lowercase letters indicate significant differences among system by entry point factors or among system by crop factors. Uppercase letters indicate significant overall differences between systems.

Earlier findings across the entire FSE established that average yields from 2002 to 2005 for corn, soybean, and wheat, but not alfalfa, were greater in CNV than ORG systems [17]. For the 2004 to 2007 growing seasons, average CNV corn and wheat yields (respectively 7.1 and 3.1 Mg ha$^{-1}$) were lower than county yield averages (Stevens County, MN; https://quickstats.nass.usda.gov/) for non-irrigated conventionally grown corn and wheat (respectively, 9.7 and 3.6 Mg ha$^{-1}$). In contrast, average CNV soybean and alfalfa yields (respectively, 2.7 and 8.6 Mg ha$^{-1}$) were slightly better than county averages (respectively, 2.6 and 8.2 Mg ha$^{-1}$). Average ORG crop yields were 42% to 60% lower than statewide organic crop yields reported in 2008 [45]. The differences between CNV and ORG yields, which ranged from 20% to 129%, were higher than expected according to results of a recent meta-analysis indicating that globally, conventional yields were 13% to 34% greater than comparable organic systems [46].

Although both total N applied and overall SMN dynamics indicated plant available N was potentially greater in ORG than CNV and should have given rise to greater crop yields, the opposite occurred. The main driver of this ORG yield deficit was high weed and insect pest pressure, as previously reported for the FSE [26]. Water availability was also an issue, particularly regarding ORG soybean production in 2006. The inability to properly control these pressures, at least during the early part of transition, points to a lack of organic management experience, recognized by Martini et al. [47]

and Seufert et al. [46] as a factor that limits organic crop yields. It also points to greater challenges in managing weed pressures in organic systems while also striving to minimize the use of intensive tillage.

### 3.2. Biomass Production and Quality

A portion of N taken up by crops, regardless of source, is ultimately removed with harvest, while the remaining N enters the residue return pool through unharvested shoot and root biomass. Total grain and shoot biomass production averaged over 2004 to 2007 was significantly greater under CNV than ORG for corn, soybean, and wheat, whereas alfalfa forage was similar (Table 4). Additionally, the HIs of conventional corn and soybean crops were greater than their organic counterparts. Total N of grain was also greater in CNV, but total N of shoot was greater in CNV than ORG only for soybean and wheat (Table 4). In contrast to above-ground production, neither biomass nor total N of roots differed between management systems. Despite the lack of differences in root biomass, significantly lower root:shoot ratios in CNV than ORG occurred in soybean and alfalfa (Table 4).

**Table 4.** Least square means[†] for annual grain, shoot, and root biomass and total N content, harvest index, and root:shoot ratio by crop under conventional (CNV) and organic (ORG) management systems over the 2004–2007 growing seasons.

| | Corn | | Soybean | | Wheat | | Alfalfa | |
|---|---|---|---|---|---|---|---|---|
| | **CNV** | **ORG** | **CNV** | **ORG** | **CNV** | **ORG** | **CNV** | **ORG** |
| Total Dry Biomass | | | | Mg ha$^{-1}$ yr$^{-1}$ | | | | |
| Grain | 7.7 [a,‡] | 2.7 [b] | 2.6 [a] | 0.7 [b] | 3.7 [a] | 1.5 [b] | – | – |
| Shoot | 6.3 [a] | 4.8 [b] | 1.9 [a] | 0.8 [b] | 4.6 [a] | 1.9 [b] | 6.8 [a] | 5.2 [a] |
| Root | 1.4 [a] | 1.3 [a] | 1.1 [a] | 0.7 [a] | 1.0 [a] | 1.1 [a] | 2.0 [a] | 1.8 [a] |
| Harvest index | 0.55 [a] | 0.37 [b] | 0.56 [a] | 0.48 [b] | 0.40 [a] | 0.41 [a] | – | – |
| Root:shoot ratio | 0.22 [a] | 0.20 [a] | 0.28 [b] | 0.79 [a] | 0.13 [a] | 0.37 [a] | 0.35 [b] | 0.68 [a] |
| Total N | | | | kg ha$^{-1}$ yr$^{-1}$ | | | | |
| Grain | 85 [a] | 38 [b] | 154 [a] | 44 [b] | 109 [a] | 41 [b] | – | – |
| Shoot | 33 [a] | 35 [a] | 16 [a] | 7 [b] | 22 [a] | 6 [b] | 210 [a] | 159 [a] |
| Root | 19 [a] | 20 [a] | 27 [a] | 16 [a] | 18 [a] | 19 [a] | 39 [a] | 32 [a] |
| Shoot + root | 52 | 55 | 43 | 23 | 40 | 25 | 50 [§] | 40 |

[†] Note: presented harvest index (HI) or root:shoot ratios do not conform to direct calculations from table data due to $e^n$ back transformation of LSMeans from a logN distribution analysis or with missing and unbalanced data; [‡] Lowercase letters within a row and crop indicate significant differences between CNV and ORG ($p < 0.05$); [§] Shoot + root for alfalfa includes only the 5% of shoot remaining in the field after harvest.

The differences observed in HI between systems were conspicuous with respect to the conventional versus organic varieties used for corn and soybean, whereas the same wheat and alfalfa varieties were used in both CNV and ORG systems (Table 1). The higher HI in CNV reflected the well-established breeding efforts for conventional crop varieties that have maximized grain production with respect to other aspects of interest such as disease or drought tolerance [48]. In contrast, breeding efforts targeting organically managed varieties are in their infancy [49]. On the other hand, similarities in HI of wheat and biomass of alfalfa between CNV than ORG probably reflected differences of in-field management rather than breeding.

The similarity of root biomass between CNV and ORG systems and among crops was consistent with previous reports for the FSE [34]. Biomass allocations to roots are predicted under nutrient limited conditions [50], but roots tend to be less responsive to N compared to shoots [51,52]. The lack of differences in root biomass implied that soil nutrient status was not sufficiently different among treatments to influence root growth. On the other hand, the differences in root:shoot ratios of soybean and alfalfa between CNV and ORG might reflect differences in N uptake processes.

Across grain crops, total grain-N was significantly greater in CNV than ORG crops ($p > 0.0001$; Table 4). This was due to greater mass, as N concentration of corn and soybean were significantly greater in ORG, at 14 ± 0.5 and 64 ± 0.5 g N kg$^{-1}$, than CNV, at 11 ± 0.5 and 59 ± 0.5 g N kg$^{-1}$, respectively ($p < 0.01$; Table S1). Total shoot-N was significantly greater in CNV than ORG for soybean and wheat due to a combination of mass and shoot N concentration, which was significantly greater

under CNV than ORG for corn and wheat (Table S1). Genetics appeared to be a factor controlling uptake and conversion of N into seed biomass for corn and soybean, which differed between the two systems. However, management probably influenced wheat differences as the same variety was planted in both systems.

A crop's ability to assimilate N and convert it into seed, shoot, and root biomass can be influenced by physiological or environmental constraints. For example, N assimilation might be impaired under low N availability conditions, as was observed in soybean by Christophe et al. [53]. A similar finding of a positive relationship of N availability and grain N content was demonstrated in wheat [54]. Alternatively, N uptake can be limited by water stress due to drought or pest pressure [53,54]. Drought stress was ruled out, as this climate factor was expected to be consistent across the experimental site with respect to soil texture and tillage influences. However, insect pests and weeds were better managed in CNV systems due to application of insecticides and herbicides [27]. The lower N concentrations in CNV crops might be explained by the dilution effect, whereby the essential amount of N needed by a crop is taken up early in growth, but N concentration is diluted later in the growing season as the plant puts on more net biomass from C fixation [55]. This effect became evident by the fact that total mass of N accumulated in corn, soybean, and wheat grain, as well as shoots of soybean and wheat, were still greater in CNV than in ORG (Table 4). Additional physiological comparisons are needed to fully understand N assimilation dynamics.

### 3.3. Residue Returns, Net Inputs, and Impact on Yields

Annual total mass, N, and C of residue returns from unharvested shoot and root biomass sources were calculated and summed within each management system-entry point combination from 2002 to 2009 (Table 5). Averaged across all entry points, total residue mass returned was 6 Mg ha$^{-1}$ higher in CNV, with 2 Mg ha$^{-1}$ more residue-C but no difference in residue-N than ORG. Among entry points, however, mass and residue-C inputs were similar between ORG-EP1 and all CNV entry points except CNV-EP4. The remaining ORG entry points were significantly lower in mass and residue-C than all other treatments. Residue-N returns were significantly greater in ORG-EP1 than all but CNV-EP3, and similar among the remaining treatments.

**Table 5.** Cumulative mass, C, and N of residue returns from 2002–2009, by conventional (CNV) and organic (ORG) system and entry point (EP).

| | CNV | | | | | ORG | | | | |
|---|---|---|---|---|---|---|---|---|---|---|
| | EP1 | EP2 | EP3 | EP4 | avg. | EP1 | EP2 | EP3 | EP4 | avg. |
| Residue returns | | | | | Mg ha$^{-1}$ | | | | | |
| Mass | 37 [a,b,†] | 38 [a,b] | 39 [a,b] | 36 [b] | 38 [A] | 43 [a] | 29 [c] | 30 [c] | 27 [c] | 32 [B] |
| C | 16 [a,b] | 16 [a,b] | 17 [a,b] | 16 [b] | 16 [A] | 19 [a] | 12 [c] | 13 [c] | 12 [c] | 14 [B] |
| N | 0.36 [c] | 0.41 [bc] | 0.46 [a,b] | 0.39 [bc] | 0.40 [A] | 0.51 [a] | 0.40 [b,c] | 0.40 [b,c] | 0.37 [b,c] | 0.42 [A] |
| C:N | 44 | 39 | 37 | 41 | 40 | 37 | 30 | 33 | 32 | 33 |

[†] Lower case letters within a row indicate significant differences among system entry points; upper case letters within a row indicate significant differences between systems.

Total nutrients from both manure and residue returns averaged over all ORG entry points exceeded those of residue returns alone in CNV by 3 Mg C ha$^{-1}$ and 50 kg N ha$^{-1}$ (Tables 2 and 5). More specifically, the observed increase in ORG corn yields (Figure S1) coincided with a 46 kg ha$^{-1}$ greater total N of inputs + returns in ORG-EP4 through 2006, respectively, 182 + 250 kg ha$^{-1}$ in ORG, compared to 100 + 286 kg ha$^{-1}$ in CNV (Table 2 and Table S2). Weed biomass would be an additional reservoir for C and N and was probably greater in ORG than CNV, given the established pressures in the FSE [26,27]. Seufert et al. [46] suggested that net N input and management experience amassing over time contributed to increased yields of organically managed crops and reduced yield deficits in comparison to conventionally managed crops. Conversely, Martini et al. [47] claimed increased yields over time after transition to organic practices were due to acquisition of management experience only, not change in nutrient availability. Over the course of the FSE, management changes were made to

improve alfalfa termination to reduce its regrowth and competitive pressure on corn. However, in the current study, management experience and N accumulation cannot be teased apart as both factors coincided with the observed yield increase in corn phases across ORG entry points.

### 3.4. Net Nitrogen Mineralization (Nmin) Dynamics and Relationship to Inputs and Yields

The capacity for the soil community to release N for plant uptake was compared across management systems using the relative measure of Nmin during a portion of the growing season. Across entry points (year random), Nmin was not significantly greater in ORG than CNV where it averaged $118 \pm 11$ and $99 \pm 11$ kg N ha$^{-1}$, respectively ($p < 0.06$). However, this difference was dependent on significant interaction with entry point ($p < 0.002$), whereby Nmin was significantly greater in ORG-EP4 ($153 \pm 13$ kg N ha$^{-1}$) than ORG-EP1 ($127 \pm 13$ kg N ha$^{-1}$), which were both greater than all other system by entry-point treatments that averaged 98 kg N ha$^{-1}$. Differences evaluated within years indicated Nmin was similar among all treatments in 2005, significantly greater in ORG-EP4 than CNV-EP3 in 2006, and greater in ORG-EP4 over all treatments in 2007 (Figure 4).

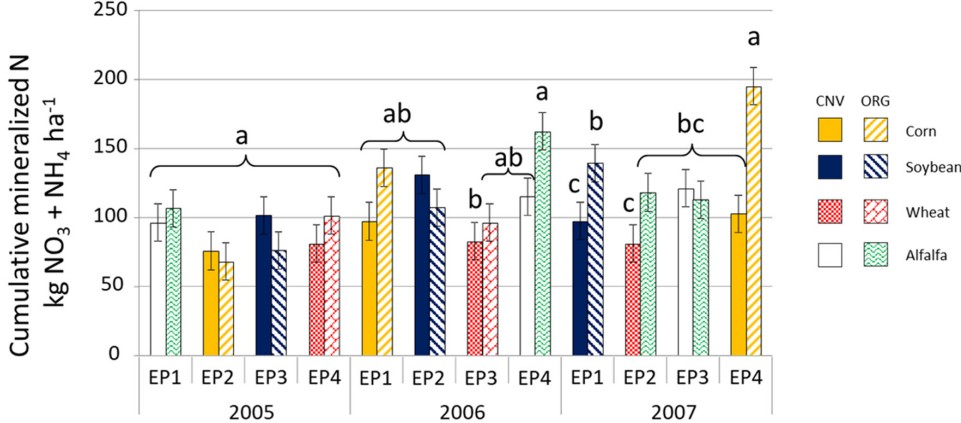

**Figure 4.** Cumulative net mineralized N (kg NO$_3^-$-N + NH$_4^+$-N ha$^{-1}$) in the 0–9 cm depth, for each crop by initial crop rotation entry point (EP) for conventional (CNV) and organic (ORG) management systems for the 2005 through 2007 growing seasons. Note: CNV wheat was under-seeded with alfalfa in all years; ORG wheat was under-seeded with alfalfa only in 2005, then dropped from the rotation in 2007. Bars show standard error of the mean. Significant differences among the eight system by entry point treatments within each year, as established by Bonferroni adjusted multiple comparison tests, are indicated by different lowercase letters ($p < 0.05$). Crop rotation EP is defined in Figure 2.

Averaged over all entry points, Nmin rate increased over time for 2005, 2006, and 2007, respectively, from 0.8 to 1.4 to 1.9 kg ha$^{-1}$ d$^{-1}$ in ORG and from 0.8 to 1.2 to 1.3 kg ha$^{-1}$ d$^{-1}$ in CNV. Broken down by entry point, Nmin-rate in the ORG notably increased each year under all entry points, but in the CNV appeared to change inconsistently (Figure 5). The increase in ORG Nmin-rate aligned with the observed increase in ORG corn yields and was consistent with the benefits of accumulating N inputs suggested by Seufert et al. [46]

The Nmin rates in ORG-EP4 was 30% to 40% greater than the overall ORG average. This was interesting given that through 2007 ORG-EP4 had the second to lowest fertilizer-N input and lowest residue-N returns, whereas ORG-EP1 had greater residue-N returns and ORG-EP3 had greater fertilizer-N inputs (Table 2, Table 5 and Table S2). These observations suggest that the recentness or the quality of inputs and residues influenced Nmin dynamics. Alfalfa appeared to be the driving factor influencing Nmin-rate dynamics. In the case of ORG-EP4, two alfalfa rotations contributed proportionally more to total residue-N before 2007 than found in the other treatments (Table S2).

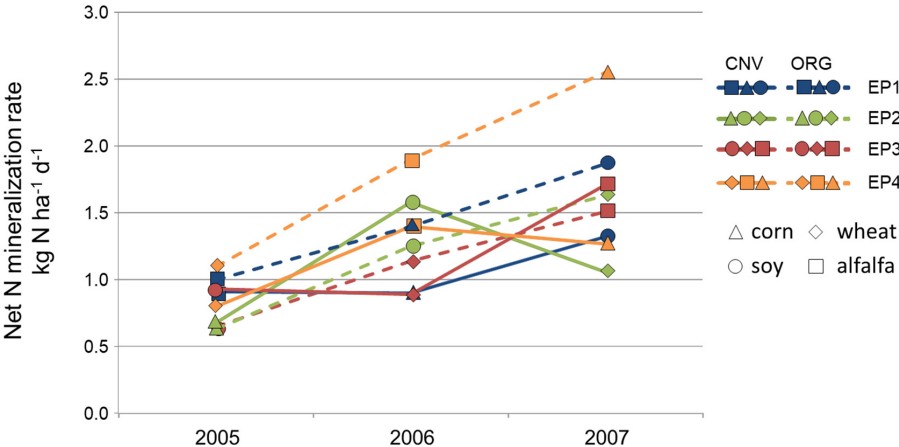

**Figure 5.** Change in net N mineralization rate (kg ha$^{-1}$ d$^{-1}$) by system and entry point over three growing seasons. Within years net N mineralization rate was obtained by linear regression on the cumulative amount of mineralized N measured over successive incubation periods.

Alfalfa was credited with enhancing the entire N balance of a production system, mainly due to the symbiotic relationship alfalfa has with rhizobia that fixes N [4]. This added N is released for use by the next crop through decomposition of residual alfalfa shoot and root biomass. For instance, a stand of alfalfa grown as a green manure, with at least 43 plants m$^{-2}$, might supply 168 kg N ha$^{-1}$ through decomposition [56]. Though not accounting for the alfalfa establishment year under wheat, the current data indicated that in a full growing season, above-ground alfalfa material might provide on average 159 kg of organically-bound N ha$^{-1}$ in a year in the ORG system; however, most of this material was removed by harvest, leaving only 40 kg N ha$^{-1}$ yr$^{-1}$ in shoot + root residue (Table 4). Moreover, decomposition rates are much higher for alfalfa than other crop residues. Specifically, Johnson et al. [57] determined laboratory-based C and N mineralization rates, averaged over root, stem, and leaf biomass, were 13% faster for active C, 92% faster for passive C, and 60% faster for N mineralization for alfalfa over corn biomass. Therefore, mineralization rates within a cropping system might be expected to be greater in the year following alfalfa production because of biomass decay.

Under CNV management, patterns in Nmin-rate were more apparent in relation to crop phases, where rates increased in CNV-EP2 and CNV-EP4 from 2005 to 2006, and in CNV-EP3 and CNV-EP1 from 2006 to 2007, when the crop phase shifted from corn to soybean, or from wheat to alfalfa (Figure 5). Conversely, Nmin rates remained stable in CNV-EP1 and CNV-EP3 from 2005 to 2006 or decreased in CNV-EP2 and CNV-EP4 from 2006 to 2007 when the crop phases shifted from soybean to wheat, or alfalfa to corn (Figure 5).

The fact that Nmin rate in CNV did not follow the same pattern of increase over time observed in ORG was puzzling, particularly with respect to the slight declines in Nmin-rate in the corn phases that followed alfalfa, despite the 25% more alfalfa shoot + root N in CNV than ORG systems (Table 4). The observed static or declining Nmin-rate paralleled the patterns observed for Nmin and SMN. Additionally, these rate patterns were associated with crop phases receiving fertilizer applications, where corn received starter fertilizer and wheat received the bulk of fertilizer applied during the 4y rotation. These observations aligned with research indicating that inorganic fertilization can depress microbial biomass [58] and activity associated with decomposition processes [59,60], i.e., mineralization of C and N from organic sources. These findings contrast with other expectations that inorganic fertilizer additions prime organic matter decomposition, and thus enhance mineralization activity [11,12].

Differences in mineralization dynamics were not explained by spring microbial biomass measurements, as microbial biomass was similar among systems and entry points from 2005 through 2007 ($p > 0.05$). Over these three years, for ORG and CNV, respectively, average MBC was 711 ± 44 and 732 ± 44 mg C kg$^{-1}$ soil, and average MBN was 127 ± 4 and 128 ± 3 mg N kg$^{-1}$ soil, across entry points.

In the FSE, microbial biomass in ST-YF-4y treatments, across entry points, was not different between CNV and ORG, but was greater compared to CT-YF-2y treatments, which suggested the overriding benefits of reduced tillage and increased crop diversity [36]. Unfortunately, the potential for differences in microbial biomass to occur during the growing season was not assessed in the current study.

Mineralization dynamics might be controlled by resource stoichiometry rather than microbial biomass [61,62]. Nutrient inputs in mineral or organic form influence the stoichiometry of the resource base that soil microbes process to either mineralize or immobilize nutrients. For example, mineralization rates will be higher at low C:N and lower at high C:N [63]. In a plant-free microcosm study, mineralized N was immobilized by the microbial community when C inputs were increased and continued to be immobilized until the addition of N reached a threshold that was no longer limiting to the microbial community [61]. This stoichiometric regulation means that for microbial processes to release N for plant uptake the microbial community cannot be N limited.

Although Nmin was measured under conditions that excluded roots, the initial amount of N in the soil, SMN, might have had an impact on microbial processes. If so, mineralization should be lower when SMN was lower and higher when SMN was higher. This appeared to be consistent only with regards to ORG corn phases, which consistently had high SMN and high Nmin. As discussed above, this was probably due to decomposition of alfalfa residue. Stoichiometrically, alfalfa had the lowest C:N and corn the highest (Table 4), thus higher mineralization rates would be expected with alfalfa decomposition and lower rates with corn residue decomposition. Comparatively, overall C:N of residue returns were lower in ORG than CNV (Table 5), but with net inputs the C:N ratio would decrease to 26 in CNV and 28 in ORG.

In summary, mineral N dynamics in CNV and ORG seemed to be regulated by different mechanisms. In the CNV, shifts between depressed or enhanced mineralization activity were aligned with specific crop phase changes, whereby in one phase mineral-N fertilizer likely depressed microbial processes or was applied at rates that were still limiting to microbial processes and, in the other, legume crops did not have to compete with soil microbes for available N. In contrast, in the ORG, an increase in corn yields was observed with an increase in mineralization activity, which indicated that the soil community was not limited by N availability. Further, dynamics in ORG treatments were able to demonstrate stochiometric controls on microbial activity. Dynamics in ORG might also have been driven by greater overall conversion of less to more available N in the form of crop and weed residues.

### 3.5. Yield N Removal, N Balance, and Implications for SOC and TN

Over the eight production years from 2002 to 2009, N accumulated in grain and forage yields averaged across entry points amounted to 1.12 and 0.77 Mg N ha$^{-1}$ in CNV and ORG, respectively (Table 6). Among entry points, total yield N exceeded the amount of total N supplied by fertilizer amendments by 3.4 to 6.0 times in CNV systems and 1.7 to 5.6 times in ORG systems. This indicated that the additional N taken up by crops must have been obtained through mineralization of SOM or crop residue, or fixation by free-living or legume-associated microbes. The gross difference of total N of inputs + returns and yield N removed resulted in a negative N balance across all CNV entry points and all ORG entry points except ORG-EP3 (Tables 5 and 6).

Negative N balances indicated potential net export of N, but soil nutrient findings indicated soil TN averaged across entry points in either system did not significantly change from 2002 to 2009 in the 0–60 cm depth profile. However, CNV-EP4 had a significant rate of loss of −0.18 Mg N ha$^{-1}$ yr$^{-1}$ and ORG-EP2 had a significant rate of gain of 0.24 Mg N ha$^{-1}$ yr$^{-1}$ (Table 7). These findings support the assumption that N fixation occurred and compensated for the apparent N deficit. Fixed N would have to account for 79% in CNV and 65% in ORG of the average total N removed in harvested material. Soybean and alfalfa together contributed an average total of 0.73 and 0.41 Mg N ha$^{-1}$ in harvested material and 0.19 and 0.28 Mg N ha$^{-1}$ in residues returned in CNV and ORG systems, from 2002 through 2009 (Table 4 and Table S2). These values indicate that legume fixation would have to account for 95 and 72% of total N of these plant materials to compensate for the average 0.88 and 0.50 Mg ha$^{-1}$

N deficit in CNV and ORG, respectively. These percentages are well within expected fixation capacities reported for soybean and alfalfa, which range upwards of 90% of total N uptake [64–66].

**Table 6.** Cumulative yield N and net N balance over 2002–2009 within the conventional (CNV) and organic (ORG) systems by entry point (EP) into the rotation.

|  | EP1 | EP2 | EP3 | EP4 | Average |
|---|---|---|---|---|---|
| Yield N |  |  | Mg ha$^{-1}$ |  |  |
| CNV | 1.25 [a, †] | 1.13 [a] | 1.12 [a] | 0.96 [a,b] | 1.12 [A] |
| ORG | 0.96 [a,b] | 0.71 [b] | 0.68 [b] | 0.73 [b] | 0.77 [B] |
| Net N Balance [‡] |  |  | Mg ha$^{-1}$ |  |  |
| CNV | −0.89 | −0.94 | −0.93 | −0.77 | −0.88 |
| ORG | −0.61 | −0.56 | −0.28 | −0.55 | −0.50 |

[†] Lower case letters indicate significant differences among system entry points; upper case letters indicate significant differences between systems. [‡] The Net N balance is the total N of net inputs minus yield N removed and does not account for N-fixation by legumes or soil N content.

**Table 7.** Intercept, slope, and *p*-value for slope of linear regressions of change in soil total N (TN) and organic C (SOC) in the 0–60 cm depth from 2002 to 2009.

| System/ | TN | | | SOC | | |
|---|---|---|---|---|---|---|
| Entry Point | Intercept | Slope | *p*-Value | Intercept | Slope | *p*-Value |
|  | Mg N ha$^{-1}$ yr$^{-1}$ | | | Mg C ha$^{-1}$ yr$^{-1}$ | | |
| CNV | 11.24 | 0.02 | 0.7150 | 145 | −2.2 | **0.0085** [†] |
| EP 1 | 11.26 | 0.03 | 0.7047 | 144 | −3.0 | **0.0150** |
| EP 2 | 11.26 | 0.19 | 0.0701 | 142 | 0.1 | 0.9100 |
| EP 3 | 11.00 | 0.06 | 0.4606 | 142 | −1.3 | 0.3430 |
| EP 4 | 11.45 | −0.18 | **0.0150** | 150 | −3.9 | **<0.0001** |
| ORG | 10.74 | 0.02 | 0.7443 | 132 | −1.2 | 0.1306 |
| EP 1 | 11.14 | −0.03 | 0.6808 | 140 | −2.4 | **0.0410** |
| EP 2 | 9.73 | 0.24 | **0.0332** | 115 | 1.3 | 0.2977 |
| EP 3 | 11.42 | −0.01 | 0.8856 | 147 | −2.0 | 0.1462 |
| EP 4 | 10.52 | 0.00 | 0.9770 | 124 | −0.8 | 0.2460 |

[†] Numbers in bold typeface highlight significant *p*-Values.

On the other hand, change in SOC from 2002 to 2009 in the 0–60 cm soil profile, averaged across entry points, was significant in CNV but not in ORG, respectively, −2.2 Mg ha$^{-1}$ yr$^{-1}$, and −1.2 Mg ha$^{-1}$ yr$^{-1}$ (Table 7; [35]). For entry points within systems, loss of SOC was significant in CNV-EP1, CNV-EP4, and ORG-EP1. Although SOC change was not significant under other entry points, significant loss might be measured in the future where negative rate trends continue. Escalated turnover and further reduction of SOC might have resulted from a decrease in the soil C:N due to loss of SOC while soil TN remained stable [35]. The observed increase in mineralization rates in ORG-EP1 might explain the loss of SOC. However, the lack of this correlation between SOC loss and mineralization in the other treatments is unclear. Potentially, SOC loss occurred in relation to the crop phase shifts affecting mineralization dynamics.

Soil N appears to have been maintained in treatments with either mineral-N or organic-N fertilizer source, except for CNV-EP4. Glendinging and Powelson [16] attributed the maintenance of soil N to use of mineral fertilizers via increased crop production and residue inputs, in comparison to a control with no fertilizer. The loss of soil TN in CNV-EP4 might be due to low productivity as indicated by the low cumulative yield N resulting from the lowest cumulative mineral-N inputs, in addition to lowest overall total N of inputs + returns. Additionally, SOC was also lost from CNV-EP4, as well as CNV-EP1 and ORG-EP1. Counter to Glendinging and Powelson [16] total N and total C of inputs + returns in these latter two treatments were the highest of all treatments, as was within-system yield N. These latter two findings reflect those of Ladha et al. [67] and Russel et al. [68] who agreed that fertilizers increased crop production and residue inputs but found these inputs did not prevent loss of SOM. Russel et al. [68]

attributed the loss of SOC to increased decay rates. However, this linkage was apparent for only one system-entry point, ORG-EP1, which had both greater Nmin rate and significant SOC loss.

## 4. Conclusions

The primary goal of this study was to provide a better understanding of management impacts on C and N dynamics to support development of effective strategies for improving intrinsic N supply in conventional and organic management systems. Better understanding was achieved by assessing yields, residue production, N mineralization, and SOM dynamics.

Greater crop yields were correlated with greater residue production. In the ORG, the entry point with greatest yield, ORG-EP1, produced greater residue inputs. Yields and residue production did not differ among CNV entry points, but both were greater than ORG. However, in contrast to hypothesis (1), the production of residues did not increase SOM under any system entry point, nor did it prevent loss of SOM, which occurred in the form of SOC in ORG-EP1 and CNV-EP1, and both SOC and TN in CNV-EP4. This indicated that strategies that currently had no significant changes might maintain a status quo, but other management improvements, particularly those that might increase overall yields, and thus residues, in both systems are needed to accrue SOM, particularly because trends were negative even if not significant.

Nutrient cycling processes differed between the two management systems as might be expected given that mineral-N inputs were greater in CNV, but total-N inputs were greater in ORG. Despite the level or form of inputs, yield N removal led to a N deficit that was greater in CNV than ORG. This deficit must have been made up by N fixation in both systems, given the lack of overall change in soil TN. Calculations indicated N fixation rates would have to be greater in CNV than ORG to compensate for the N deficit, however total N of residues from legumes was greater in ORG. In addition, weed biomass and organic-N of manure contributed to larger pool size and different composition of these organic matter resources in ORG then CNV. This could explain why SMN and Nmin, as measures of N availability, were generally greater in ORG than CNV. In contrast to hypothesis (2), SOM was not increased in either management system, but the different N cycling dynamics in ORG, which were driven by resource stoichiometry, indicated this system had a greater capacity to provide N through intrinsic sources. These results suggest further research is needed to evaluate the potential of external inputs of C and N, as well as inclusion of legumes to improve soil C and N status regardless of system level management.

**Supplementary Materials:** The following are available online at http://www.mdpi.com/2073-4395/10/6/764/s1, Figure S1: Average annual corn, soybean and wheat grain and alfalfa forage yields (Mg ha$^{-1}$) harvested from conventional (CNV) and organic (ORG) management systems, over all entry points from 2002 to 2009, Table S1: Crop grain, shoot and root biomass N concentration (g kg$^{-1}$) for corn, soybean, wheat and alfalfa in conventional (CNV) and organic (ORG) management systems sampled from 2004 through 2007, Table S2: Yearly and cumulative sum of total residue-N inputs (kg N ha$^{-1}$) by entry points within conventional (CNV) and organic (ORG) management systems from 2002 to 2006 and 2002 to 2009.

**Author Contributions:** Experimental design, D.W.A.; Conceptualization, analysis and writing—original draft preparation, S.L.W.; investigation, S.L.W., D.W.A., J.M.F.J., A.R.W.; writing—review and editing, D.W.A., J.M.F.J., A.R.W.; All authors have read and agreed to the published version of the manuscript.

**Funding:** This research was supported by USDA-ARS Project Number: 5060-11610-003-00-D.

**Acknowledgments:** We thank N. Barbour, J. Eklund, C. Hennen, S. Larson, D. Peterson, S. VanKempen, and J. Hanson for their technical assistance in plot maintenance, microbial biomass sample collection, and sample analysis. We also thank the reviewers for their help in improving this manuscript. This work contributes to USDA-ARS GRACEnet program. The dataset is available at: https://data.nal.usda.gov/dataset/farming-systems-study-greenhouse-gas-reduction-through-agricultural-carbon-enhancement-network-morris-minnesota. Mention of trade names or commercial products in this publication is solely for the purpose of providing specific information and does not imply recommendation or endorsement by the US Department of Agriculture. USDA is an equal opportunity employer and provider.

**Conflicts of Interest:** The authors declare no conflict of interest.

**Appendix A**

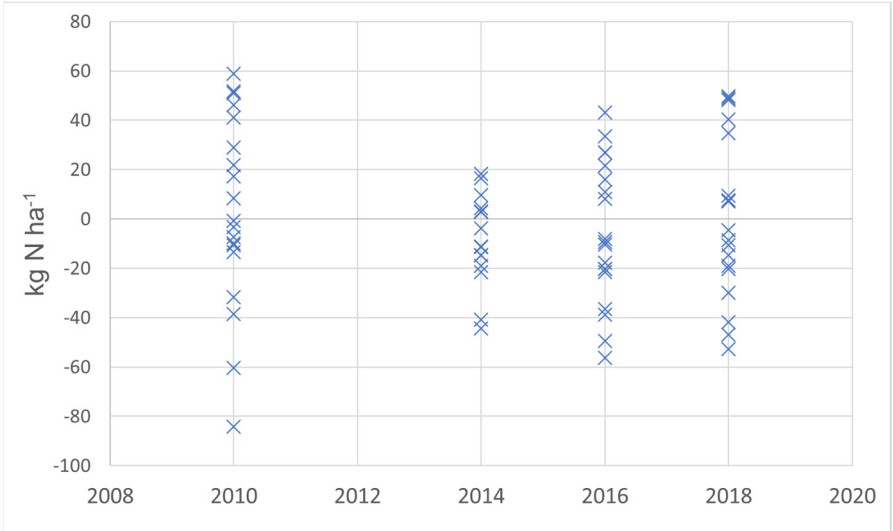

**Figure A1.** Range of differences in fertilizer N applied and estimated corn grain N removed for four years of available data from 19 reporting US states [5].

Survey data from the USDA National Agricultural Statistics Service for 2010, 2014, 2016 and 2018 were accessed using the "Quickstats" application (http://quickstats.nass.usda.gov/). A total of 19 states provided both state-wide average yields (bushels per acre) and fertilizer application rates (pounds of N per acre). These values were converted into kg ha$^{-1}$. We then calculated the potential N removal rate in the grain using a standard concentration of N of 0.016 kg N kg$^{-1}$ grain (0.90 lb N bu$^{-1}$) suggested by Murrell [69]. This value matches with concentrations reported in the current study (Table S1), and in other reports [70,71], but might also result in poor estimates [68]. We then subtracted the calculated grain-N in kg ha$^{-1}$ from fertilizer-N applied in kg ha$^{-1}$. Positive values indicate fertilizer N excess, negative values fertilizer N deficits. Although many states oscillated each year between excess and deficit (not shown), typically more than 7 of 19 states demonstrated excess N (Figure A1).

**Appendix B**

The following is a brief synopsis of methods for measuring plant and soil parameters that are provided in complete form elsewhere. Aboveground vegetative shoot material (stover, stubble, and straw) was collected from a 1-m$^2$ area in each plot after peak maturity of corn, soybean, and wheat. Alfalfa was sampled from a 1-m$^2$ area at each of three harvest times a year. Root material was sampled at peak maturity (75% silk in corn, 1st pod in soybean, 75% boot in wheat, and third harvest in alfalfa) by compositing twelve 2-cm soil cores taken to a 60-cm depth in an arrangement to account for root distribution. Plant materials, including grain, were dried at 45 °C, fine ground, and analyzed for C and N by direct combustion (Leco Tru-Spec CN Analyzer, Leco Corp, St. Joseph, MI, USA). Additional details are provided by Johnson et al., [33,34].

Soil samples were taken in the fall in 2002, 2003, 2007, and 2009 using a tractor mounted probe. Two soil cores of 6.5 cm to at least 60cm depth were divided at 0–5, 5–10, 10–15, 15–30, and 30–60 cm. One core was used for bulk density and one for chemical analysis after drying at 37 °C, finely ground, and analyzed for C and N by direct combustion as above. Soil organic C and N contents were evaluated on an equivalent mass basis [35]. Additional soil samples were taken every spring from 2002 through 2010, by compositing three randomly taken soil cores 2.5-cm diameter and 15-cm depth. Duplicated samples were either directly extracted or extracted following the chloroform fumigation-direct extraction procedure of Vance et al. [72] using 0.5 M K$_2$SO$_4$, and analyzed for total N

and organic C using an IL550 TOC/TN analyzer (Hach-Lange). Further details on soil analyses are provided by Weyers et al. [35,36].

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
