# Peer review of "Management Drives Differences in Nutrient Dynamics in Conventional and Organic Four-Year Crop Rotation Systems"

_agronomy, doi:10.3390/agronomy10060764_

Round 1

Reviewer 1 Report

The soil analyses should be given for each soil type, different soils are able to contribute nutrients at different levels, particularly those that have a high clay component (i.e. K). 

The 'average' P levels are on the low side - were these adjusted at the beginning of the trial?

The comparison between organic and conventional nutrient loading would require equal amounts of N, P and K for each crop for each system. The nutrients e.g. nitrogen are partially available and partially unavailable in the organic form, therefore if a fair comparison is to be made the conventional mineral fertiliser application and organic forms of nutrients,  the total nutrients (N, P, K and Mg at a minimum but should also include secondary and micronutrients) in the manure should equal the nutrients applied in the mineral fertiliser.  The analysis of the manure should inform the nutrient application on the conventional rotation. The hypothesis' are invalidated by the fact that the nutrient applications in the two systems is different. Yields cannot be compared between the two systems because the rates of nutrient application are different.

Author Response

Comments and Suggestions for Authors

The soil analyses should be given for each soil type, different soils are able to contribute nutrients at different levels, particularly those that have a high clay component (i.e. K).

We appreciate your comment and we recognized that crops can respond to differences in soil types.    Our statistical assessment of soil properties (Jarardat and Weyers 2011 [20]], cited on LINES 64-65, indicated that variability is addressed by the experimental plot layout, such that differences in soil properties, i.e., nutrient content, CEC, texture, are normalized, or insignificant.  As we did not evaluate the 32 experimental plots used in the current study with respect to the distributions in soil type distributions, we feel that it would be unnecessary to provide this data for each soil. 

The 'average' P levels are on the low side - were these adjusted at the beginning of the trial?

In our experience with this area of investigation, P deficiency for crop productivity was never observed.  Soils in this area are erosive and nature and state extension guidelines often recommend limitation on P amendments to crop needs, not soil availability.  Our fertilization regime considers these recommendations and we applied the P fertilization levels necessary for crop production. 

The comparison between organic and conventional nutrient loading would require equal amounts of N, P and K for each crop for each system. The nutrients e.g. nitrogen are partially available and partially unavailable in the organic form, therefore if a fair comparison is to be made the conventional mineral fertiliser application and organic forms of nutrients,  the total nutrients (N, P, K and Mg at a minimum but should also include secondary and micronutrients) in the manure should equal the nutrients applied in the mineral fertiliser.  The analysis of the manure should inform the nutrient application on the conventional rotation. The hypothesis' are invalidated by the fact that the nutrient applications in the two systems is different. Yields cannot be compared between the two systems because the rates of nutrient application are different.

Thank you for this comment.  We have received similar criticism on earlier drafts of this manuscript and in response have added more data to track nitrogen addition and cycling.  The problem with trying to normalize the fertilization scheme between a mineral-N based fertilizer, and an organic-N based fertilizer that contains both mineral and organic N forms, is which form to normalize the application to.  If mineral N in the organic manure was used, an under application of N in the CNV would be likely. If organic N or total N were to be used, more available N, or an over application, would be applied to the CNV, some of it likely to be lost. 

We have responded to your comment and those of the other reviewers by shifting our focus away from a fertilizer-effect to a management system-effect.  Our hypotheses as originally stated LINES 57-58:  The hypotheses tested were 1) SOM will be improved in treatments with greater crop production and residue returns, 2) increased N availability will be associated with improvements in SOM.), were revised to: “The hypotheses tested were 1) greater crop production and residue returns will improve SOM, 2) improvements in SOM will be associated with increased N availability from intrinsic N sources.  

The title of the manuscript was also changed to focus on management aspects.  NEW TITLE: Management drives differences in nutrient dynamics in conventional and organic four-year crop rotation systems

We did not feel that these hypotheses specifically addressed any difference in fertilizer amounts to these two management systems, but whether the ability of the nutrient cycling dynamics within these management systems would be able to convert available N into yield, thus residue biomass and thus have an impact on SOM content and generation of intrinsic N.  The introduction material presents a goal that both management systems might benefit by generating internal/intrinsic N sources. Our study, through comparison and contrast, improves our understanding how the management strategies within both systems influence the production or loss of SOM and generation of intrinsic N.  One of these important contrasts is made with the yield and residue production data.  We expect that yields would be different, but what we also hoped to find was that the greater production of residues, that we link to yield, would be associated with SOM increases.  We didn’t find that, but the contrast allowed us to better show that when manure, or an addition of organic matter is made, nutrient cycling processes are benefited through numerous mechanisms, one of them also being inputs from legume biomass. 

We have made numerous revisions to the manuscript to improve our communication of this concept. 

Reviewer 2 Report

General and specific comments can be found in the attached document.

Reviewer 3 Report

May I congratulations the authors for addressing significant and important agronomic topic to farmers, agronomist, crop consultants, and soil scientists. The economics of fertilizer, its application and nutrient dynamics in the soil, and its impact on farmers' profitability make its a topic worth studying scientifically, which makes this study very appropriate.

The introduction to the manuscript was very well written and well referenced to set the tone for the subsequent section in the manuscript. The research hypotheses were clear and appropriate statistical methods were used to analyze the research data to address the hypotheses. 

The materials and methods section was very well written and also well referenced with an excellent site description and appropriate experimental design. 

The management practices, soil sampling and analysis sections were also very well written and referenced.

An appropriate statistical model was used to analyze the research data, which authenticates/validates the research results.

The results and discussion section was very well written, appropriately referenced and very informative. Section 3.4 "Net nitrogen mineralization (N min) dynamics and relationship to inputs an yields" was fun to read. 

Figures and Tables in the manuscript are consistent with topics addressed in the various sections of the manuscript to appropriately discuss and explain the results of the study.

The conclusions were appropriate, well written and address the hypotheses of the study.

Overall, the study is scientific and original: appropriately designed, data collection methods were scientific, and an appropriate statistical model used to analyze the data. The manuscript was very well written by the authors.

I congratulate the authors for an excellent contribution to the field of agronomy and soil science.

Author Response

We are humbled by your gracious approval of our manuscript. Thank you for finding it of value and learning from our efforts. 

Round 2

Author Response

I am very thankful for your attention to our manuscript and the comment and suggestions that you provided to help us make it better.

Response to your second review are provided as an attachment.
